# The P2Y12 Receptor Antagonist Selatogrel Dissolves Preformed Platelet Thrombi In Vivo

**DOI:** 10.3390/jcm10225349

**Published:** 2021-11-17

**Authors:** Lydie Crescence, Markus Kramberg, Martine Baumann, Markus Rey, Sebastien Roux, Laurence Panicot-Dubois, Christophe Dubois, Markus A. Riederer

**Affiliations:** 1Aix Marseille Université, INSERM 1263, INRAE 1260, C2VN, 27 Boulevard Jean Moulin, 13385 Marseille, France; lydie.crescence@univ-amu.fr (L.C.); laurence.panicot-dubois@univ-amu.fr (L.P.-D.); christophe.dubois@univ-amu.fr (C.D.); 2Drug Discovery Biology, Idorsia Pharmaceuticals Ltd., CH-4123 Allschwil, Switzerland; markus.kramberg@idorsia.com (M.K.); mariemartine.baumann@gmail.com (M.B.); markus.rey@idorsia.com (M.R.); sebastien.roux@idorsia.com (S.R.)

**Keywords:** P2Y12 receptor, platelets, thrombosis, haemostasis, thrombus dissolution

## Abstract

Selatogrel, a potent and reversible antagonist of the P2Y12 receptor, inhibited FeCl_3_-induced thrombosis in rats. Here, we report the anti-thrombotic effect of selatogrel after subcutaneous applications in guinea pigs and mice. Selatogrel inhibited platelet function only 10 min after subcutaneous application in mice. In addition, in a modified Folts thrombosis model in guinea pigs, selatogrel prevented a decrease in blood-flow, indicative of the inhibition of ongoing thrombosis, approximately 10 min after subcutaneous injection. Selatogrel fully normalised blood flow; therefore, we speculate that it may not only prevent, but also dissolve, platelet thrombi. Thrombus dissolution was investigated using real-time intravital microscopy in mice. The infusion of selatogrel during ongoing platelet thrombus formation stopped growth and induced the dissolution of the preformed platelet thrombus. In addition, platelet-rich thrombi were given 30 min to consolidate in vivo. The infusion of selatogrel dissolved the preformed and consolidated platelet thrombi. Dissolution was limited to the disintegration of the occluding part of the platelet thrombi, leaving small mural platelet aggregates to seal the blood vessel. Therefore, our experiments uncovered a novel advantage of selatogrel: the dissolution of pre-formed thrombi without the disintegration of haemostatic seals, suggesting a bipartite benefit of the early application of selatogrel in patients with acute thrombosis.

## 1. Introduction

Pathological imbalances of the regulatory mechanisms of haemostasis enhance the probability for the formation of occlusive platelet-rich thrombi, causing major adverse cardiac events (MACEs) [1]. ADP-mediated activation of the P2Y12 receptor has been reported to be essential for the amplification of platelet aggregation, and, thus, thrombus formation [2,3]. Consequently, chronic inhibition of the P2Y12 receptor was shown to prevent MACEs in multiple clinical studies, leading to treatment guidelines for P2Y12 receptor antagonism in patients with acute coronary syndrome [4,5,6]. Coronary thrombosis is a dynamic process, with platelet thrombi being formed and broken down, a process that can take hours or even days to result in symptomatic ischemic occlusion [7]. Therefore, acute administration of a rapid-onset P2Y12 receptor antagonist is particularly relevant in the context of early thrombus formation in acute myocardial infarction. Such an approach represents an additional option to be investigated with the possibility of preventing early thrombotic propagation [8,9].

In vitro studies have indicated that the P2Y12 receptor induces intraplatelet signalling and results in stabilisation of the active conformation of the fibrinogen receptor GPIIb/IIIa [10]. Multiple studies have described the intraplatelet signalling cascade involved in the activation of the GPIIb/IIIa receptor [10].

Thus far, the experimental profiling of P2Y12 receptor antagonists has focused on a preventative effect, i.e., P2Y12 receptor antagonists being applied first; subsequently, a thrombotic insult is induced. However, information about the effect of P2Y12 receptor antagonists on pre-existing platelet-rich thrombi is scarce. For example, the perfusion of human blood over a prothrombotic surface generates thrombus formation ex vivo [11]. In this model, the addition of the P2Y12 receptor antagonists 2MeSAMP or AR-C69931MX to a growing thrombus blocked thrombus growth and induced dissolution of the platelet-rich thrombus [11,12,13]. However, performance of the ex vivo perfusion system necessitates that human blood is anticoagulated to prevent spontaneous platelet activation prior to initiation of the experiments. Citrate anticoagulation reduces the calcium concentration to non-physiological levels, limiting coagulation but also affecting platelet function [14,15,16]. However, secondary haemostasis, defined by fibrin formation, is responsible for the stabilisation of platelet aggregates [17,18]. Therefore, the described thrombus dissolution ex vivo might be facilitated by the absence of fibrin formation or by the impact of non-physiological calcium concentrations.

Selatogrel (ACT-246475) is a potent and reversible antagonist of the P2Y12 receptor [19]. In an experimental thrombosis/haemostasis model in rats, at a comparable level of antithrombotic effects, selatogrel reduced blood loss more than clopidogrel or ticagrelor [19,20,21].

The aims of our study were (i) to characterise the rapid absorption of selatogrel after subcutaneous application in mice; (ii) to characterise the antithrombotic effects of selatogrel, in the absence of anticoagulation, in experimental in vivo thrombosis models in mice and guinea pigs; and (iii) to investigate the potential effect of selatogrel on thrombus dissolution in vivo.

## 2. Materials and Methods

Reagents: Selatogrel (ACT-246475) was used in the form of its hydrochloride salt (batch ELN095-0829.1) and was produced by Idorsia Pharmaceuticasl Ltd., Allschwil, Switzerland.). Alexa Fluor 488 (AF488)-fibrinogen was purchased from Thermo Fisher, Reinach, BL, Switzerland and handled as described on the provider’s data sheet. The anti-fibrin antibody was a generous gift from Prof. Bruce Furie, Harvard Medical School, and was produced and conjugated in-house with Alexa Fluor 647 (Emax 667 nm). X488 (Emax 519 nm) was used to label platelets in in vivo experiments and was from Emfret.

Animals: WT C57BL/6J mice were used for real-time intravital microscopy experiments and were obtained from the Janvier Laboratory (Le Genest-Saint-Isle, France). All animal care and experimental procedures were performed as recommended by the European Community Guidelines (directive 2010/63/UE) and approved by the Marseille Ethical Committee #14 (protocol number: 27-28092012 and APAFIS number: 15334-2018060115491816V2).

Female Balb/c mice were used for s.c. selatogrel application and were purchased from Charles River laboratories (Sulzfeld, Germany). Normotensive male Dunkin Hartley guinea pigs were obtained from Envigo (Venray, The Netherlands) and were group-housed in climate-controlled conditions with a 12 h light/dark cycle, in accordance with the guidelines of the Basel Cantonal Veterinary Office (license no. 170).

Blood Sampling with Mice: After an acclimation period of at least 7 days, female Balb/c mice were anesthetised with isoflurane (2–5%) (Animal license #170, Kanton Baselland, Switzerland). The animals were then placed on a thermostatically controlled heating table to maintain a body temperature between 36 and 38 °C. A polyurethane catheter (BTPU-027) was inserted into the right jugular vein and advanced into the vena cava for the infusion of heparin and to collect blood. In order to perform accurate platelet assays, it was necessary to ensure that blood collection did not lead to platelet activation. Therefore, unfractionated heparin (4000 U/kg/2.5 mL) was intravenously injected to prevent blood from coagulating. Two minutes after the injection of heparin, 60 µL of whole blood was collected via the vena cava catheter (baseline). After baseline blood sampling, the test compounds were subcutaneously injected into the lower abdomen of the animals. At times of 10, 20, 30, 40 and 60 min after subcutaneous injection of the test compounds, 60 µL of whole blood was collected via the vena cava catheter for the fibrinogen binding assay.

Ex Vivo Fibrinogen Binding Assay: AF488–fibrinogen binding to mouse platelets was quantified with samples of 6 µL of whole blood. A volume of 10 µL of AF488–fibrinogen (0.5 mg/mL final concentration) was added and incubated for 5 min. AF488–fibrinogen binding was induced by the addition of 10 µL ADP (20 µM final concentration) and incubated for 5 min. In addition, incubation with a platelet-specific allophycocyanin (APC) anti-mouse CD61 (integrin beta 3) antibody (clone 2C9.G2) was performed for 10 min. The platelets were then fixed by the addition of 34 µL of cold paraformaldehyde solution (1% final concentration) and stored at 4 °C for approximately 20 min. The samples were finally diluted with 1.8 mL of cold PBS and stored at 4 °C for at least 2 h. Platelets were first gated using the platelet-specific fluorescence signal of APC-CD61 (APC settings for fluorescence emission at 660 nm). The AF488–fibrinogen signal (FITC settings for the fluorescence emission at 525 nm) was quantitated as the mean fluorescence intensity (MFI) of 5000 platelets.

Modified Folts Model in Guinea Pigs: After an acclimation period of at least 7 days, the guinea pigs were anesthetised by an intramuscular injection of 60 mg/kg ketamine (Ketanarkon, Streuli Pharma AG, Uznach, Switzerland) and 8 mg/kg xylazine (Xylazin, Streuli Pharma AG, Uznach, Switzerland). The animals were then placed on a thermostatically controlled heating table to maintain body temperature at between 36–38 °C. After tracheotomy, the right carotid artery was gently dissected free from connective tissue, and a silastic tubing flow transducer (D-20–0.8 mm, Triton Technologies Inc., Arizona, USA) was placed on the artery for measurement of the blood flow velocity. Two millimetres upstream of the Doppler flow probe, damage to the subendothelium was induced by pinching a 1 mm segment of the dissected carotid artery with forceps (BD312R, Aesculap, Tuttlingen, Germany) for 5 s to induce thrombus formation. After damage, the carotid blood velocity started to decline until complete occlusion. When the blood flow velocity approached zero, gentle shaking of the carotid artery broke the occlusive thrombus loose and restored the blood flow at cyclic intervals. After obtaining 3–6 stable cyclic flow variations (CFVs), vehicle (mannitol 5%) or selatogrel (30 µg/kg) was administered in a blinded fashion as a subcutaneous bolus injection at a volume of 1 mL/kg. The anti-thrombotic dose of selatogrel was previously determined in a dose–response experiment (data not shown). The ongoing cyclic thrombotic process was quantified by the number of CFVs over a 30-min observation period. Carotid blood flow velocity was continuously recorded on a PowerLab data acquisition system (IOX Data acquisition, Emka Technologies, Paris, France) using IOX software (Emka Technologies, Paris, France). Data were exported to a Dell OptiPlex 7050 computer for analysis (Datanalyst version 2.6.1.12, Emka Technologies, Paris, France). At the end of the experiment, the guinea pigs were euthanised by the intracardial injection of pentobarbital (100 mg/kg; Esconarkon, Streuli Pharma AG, Switzerland).

Intravital Microscopy: Intravital video microscopy of the cremaster muscle microcirculation was performed as previously described [22]. Briefly, mice were anesthetised with an atropine (0.25 mg/kg), xylazine (12.5 mg/kg), ketamine (125 mg/kg) mixture by intraperitoneal injection. Mice were placed in a supine position, and an incision was instigated in the skin at the level of the submaxillary glands. The glands were cleared to access the jugular vein. The jugular vein was cannulated to maintain the anaesthesia of the animal with thiopental (25 mg/kg) and to inject antibodies necessary for the experiment. A cannula was also placed on the trachea to facilitate breathing during the experiment. Then, the cremaster muscle was exposed, and superfused with a saline solution and pine on a coverslip, as previously described [21]. Microvessel data were obtained using an Olympus AX microscope with a 60 × 0.9 NA water-immersion objective. Digital images were captured with a Cooke Sensicam CCD camera in 640 × 480 pixel format matched with an fluorescent signal intensifier (Intelligent Imaging Innovations) [22].

Antibodies directed against fibrin (Fib-Alexa Fluor 647 homemade) and X488 directed against platelets (Emfret) were infused through the jugular vein into the circulation of anesthetised mice before performing a laser injury [23]. Vessel wall injury was induced with a nitrogen dye laser (Micropoint; Photonics Instruments) focused through the microscope objective, parafocal with the focal plane, and aimed at the vessel wall. Image analysis was performed using Slidebook (Intelligent Imaging Innovations, Denver, CO, USA). Fluorescence data were captured and digitally analysed as previously described to determine the median fluorescence intensity signal over time and the area under the curve.

Two settings were used to study the dissolution of the newly formed thrombus. In the acute setting, selatogrel was infused through the jugular vein 62 s after the laser-induced injury and observed for 340 s. Kinetics of wild-type thrombi were obtained in mice injected with NaCl 0.09%.

In the consolidated setting, after the initial thrombus was recorded for 1400 frames (300 s) after the laser-induced injury (T0), the thrombi were allowed to consolidate in vivo for an additional 30 min. Then, the thrombus was recorded for 300 s (T30 min). Next, selatogrel was injected, and the thrombus was again recorded for 300 s (T35 min). The selatogrel dose was previously shown to fully inhibit laser-induced thrombosis [21]. Finally, 30 min after the injection of selatogrel, the thrombus was recorded for 300 s (T65 min). In this experimental setting, only one thrombus was performed per mouse. Digital images were analysed with Slidebook software (Intelligent Imaging Innovations).

Human blood sampling and light transmission aggregation: Blood sampling from healthy volunteers was approved by the local ethical committee (“Ethik Kommission Nordwest- und Zentralschweiz” [EKNZ.ch], Project 249/02)”. After obtaining informed consent, blood was collected by venepuncture of the large arm vein from healthy, aspirin-free volunteers. The blood was immediately anticoagulated with napsagatran (100 µM final). Platelet-rich plasma (PRP) and platelet-poor plasma (PPP) were obtained through differential centrifugation. Platelet numbers were determined using a Casy cell counter system (OMNI Life Science (60 µM capillary)), and the platelet count was adjusted with the corresponding PPP to obtain a final platelet concentration of 250 × 106 Plts/mL.

Light transmission aggregation (LTA) was performed according to Born using a four-channel aggregometer (Chrono-Log Corporation, Havertown, PA, USA, Lumi-Aggregometer 490-4D) with the AggroLink software package [24]. LTA was performed in a siliconized glass cuvette from Probe and Go (Lemgo, Germany). Aggregation was initiated by the addition of ADP (20 µM), and samples were stirred with siliconized magnetic stir bars at 1100 rpm. The results are expressed as the maximal change in light transmission at the peak response (percentage extent).

**Statistics:** Significance was determined with the Wilcoxon rank-sum test for the in vivo experiments and Student’s t-test for the in vitro experiments. Differences were considered significant at *p* < 0.05.

## 3. Results

### 3.1. Selatogrel Is Rapidly Absorbed after Subcutaneous Injection in Mice

The potential for rapid uptake after the subcutaneous injection of selatogrel was first evaluated in mice. To enable multiple blood samplings from the same mouse in combination with a platelet function test in whole blood, a novel method was established. To prevent the spontaneous ex vivo coagulation of blood during sampling and sample processing, unfractionated heparin was intravenously infused 2 min prior to the injection of selatogrel. Selatogrel was injected subcutaneously into the lower abdominal skin of the mouse, and 60 μL samples of blood were collected via the vena cava catheter at several time points. The presence of antithrombotic concentrations of selatogrel was assessed by a functional platelet assay where the binding of fluorescently labelled fibrinogen (AF488–fibrinogen) to platelets was quantified by FACS analysis (mean fluorescence intensity (MFI)). Stimulation with 10 μM ADP resulted in an average MFI of 6415 ± 167 (Figure 1A, ADP, open bar). The addition of 3 μM selatogrel prior to activation with ADP reduced fibrinogen binding to an average MFI of 319 ± 41 (Figure 1A, ADP + Selatogrel, black bar). In agreement with data reported in the literature [10,25,26], in non-stimulated platelets, constitutive P2Y12 receptor signalling also supported the baseline binding of AF488–fibrinogen, resulting in an average MFI of 1253 ± 193 (Figure 1A, Control, grey bar). The addition of 3 μM selatogrel reduced the average MFI to 336 ± 19 (Figure 1A. Control + Selatogrel, checked bar), confirming that antagonism of the P2Y12 receptor translates to inactivation of the GPIIb/IIIa receptor, thus triggering the release of pre-bound fibrinogen. As a next step, blood samples (6 μL blood) from various time-points after the s.c. application of selatogrel were activated by the addition of ADP (20 μM), and AF488–fibrinogen binding was quantified. A reduction in platelet-associated fluorescence was indicative of interference with fibrinogen binding, and, thus, antagonism of the P2Y12 receptor. Selatogrel was demonstrated to reduce ≥60% of fibrinogen binding only 10 min after subcutaneous application in mice (Figure 1B).

### 3.2. Subcutaneous Injection of Selatogrel Is Antithrombotic in Guinea Pigs

To confirm the antithrombotic effect of selatogrel after s.c. application, a modified Folts thrombosis model was applied in guinea pigs [27]. In this model, thrombus formation was initiated by mechanical damage of the carotid artery, and the process of thrombus growth, up to full blood vessel occlusion, was quantified by monitoring the blood flow. By mechanically embolising the thrombi, the process could be monitored cyclically (cyclic flow variation (CFV)), as previously reported [27]. In the first phase, after an initial injury (Figure 2, white arrow), multiple CFVs were induced (Figure 2, black tracing). In the second phase, selatogrel was injected subcutaneously (Figure 2, black arrow). Subsequent tracings are depicted in grey. Approximately 10 min after the injection of selatogrel, ongoing CFVs were fully blocked. In addition, selatogrel stopped the decrease in blood flow, which is indicative of the inhibition of platelet thrombus formation (Figure 2, green arrow). Furthermore, as indicated by the red arrows in Figure 2, blood flow increased back to baseline levels, suggesting that selatogrel dissolved the existing platelet thrombus. Increases in the blood flow were gradual and not abrupt, suggesting a detachment of single platelets or very small aggregates. In this experimental setting, the detachment of platelets did not cause an obvious occlusion of blood vessels downstream of the dissolved thrombus.

### 3.3. Selatogrel Dissolves Platelet Thrombi in Mice

The effect of selatogrel on thrombus dissolution was further investigated in mice using real-time intravital microscopy. In a previous study, the signal resolution provided by intravital microscopy in combination with fluorescent labelled platelets could be utilised to differentiate between thrombus embolisation and the detachment of single/a few platelets [21]. Therefore, we investigated the effect of selatogrel, applied through intravenous infusion, on laser-induced thrombosis in mice [21]. In the absence of a platelet inhibitor, platelet thrombus formation peaked, on average, 70–100 s post-laser injury, followed by a spontaneous reduction in thrombus size (Figure 3, Panel A, black tracing).

To investigate the effect of selatogrel on a pre-formed platelet-rich thrombus, the infusion of selatogrel was initiated 62 s after laser injury (Figure 3, Panel A, grey arrow). The effects of selatogrel began approximately 60 s after the start of infusion, indicated by an accelerated reduction in thrombus size relative to the control (Figure 3, Panel A, grey tracing). These data agree with previously observed thrombus dissolution patterns in guinea pigs, confirming that selatogrel not only blunts thrombus growth, but also destabilises newly formed, platelet-rich thrombi in vivo. In addition, these data confirmed that mechanical dislodgement applied in the Folts model does not facilitate the process of selatogrel-mediated thrombus dissolution.

Secondary haemostasis is defined as the fibrin-mediated stabilisation of platelet thrombi functioning as haemostatic seals [17,18]. To confirm that this effect is not mediated by interference with secondary haemostasis, fibrin formation was quantified and expressed as the relative ratio of fibrin to platelets. As presented in Figure 3, Panel B, fibrin formation, quantified at 100, 200, and 300 s after laser injury, increased over time in the absence (black) or presence of selatogrel (grey bars). In the presence of selatogrel, a more pronounced increase in the fibrin/platelet ratio was observed. This effect is in agreement with the selatogrel-mediated reduction in platelet thrombus size, and, thus, the relative increase in the ratio of fibrin per platelet, confirming that the presence of selatogrel does not interfere with the process of fibrin formation. Nevertheless, despite the presence of fibrin, selatogrel induced the dissolution of newly formed platelet thrombi (Figure 3A), confirming that P2Y12 receptor antagonism translates into disruption of the interaction between the GPIIb/IIIa receptor and fibrin.

### 3.4. Selatogrel Dissolves Consolidated Platelet Thrombi

To investigate whether selatogrel can also dissolve platelet-rich thrombi which were incubated to consolidate in vivo, the experimental procedure was adapted. Laser-induced formation of platelet-rich thrombi was performed (T0) in the absence of a platelet antagonist. No experimental manipulation occurred for the subsequent 30 min to allow blood flow over the platelet thrombus and consolidation of the platelet-rich thrombus. As a first step, we confirmed that the thrombus size, in the absence of a P2Y12 receptor antagonist, remained constant during the time window between 30–65 min after the laser-induced injury (Figure 4, Panel A and B). Longer periods of thrombus consolidation were limited due to animal welfare regulations.

In the second step, selatogrel was infused after 30 min of thrombus consolidation in vivo (T30). As presented in Figure 4, Panel C, selatogrel was able to rapidly dissolve the consolidated platelet-rich thrombi (Figure 4, Panel C, T35 and T65). The fluorescence signals associated with the mural platelet thrombi were quantified and are presented in Figure 4, Panel D. In agreement with a previous report by Crescence at al. [21], selatogrel (1 μg/kg i.v.) reduced thrombus size but did not fully dissolve mural platelet aggregates at the site of laser injury (Figure 4, Panel C, T35 and T65).

### 3.5. Selatogrel Dissolves Aggregated Human Platelets

To translate the above observations to human platelets, the dissolution of pre-aggregated human platelets was monitored in vitro in light transmission aggregation (LTA) experiments. In LTA induced by ADP (20 μM) (black open arrow), in the absence of a platelet inhibitor, platelet aggregation peaked approximately 3 min after the addition of ADP with a maximal extent of 80% aggregation after 4 min (Figure 5, black tracing, black arrow, Control). In the time window from 4–24 min, light transmission decreased to a level equivalent to 60% aggregation. To investigate the effect of selatogrel, the concentration equivalent to Cmax in human patients was chosen [28]. Aggregation was initiated by ADP (red open arrow), and selatogrel (1 μM) was added at the point of maximal aggregation (red arrow). Selatogrel induced an accelerated decrease in light transmission within the next 4 min down to 3% aggregation, suggesting rapid and complete disaggregation of the pre-aggregated platelet thrombi (Figure 5, red tracing). This observation confirmed that, in human platelets, interference with the P2Y12-receptor-mediated intraplatelet signalling destabilises the active conformation of the GPIIb/IIIa receptor, translating into the disaggregation of platelets.

## 4. Discussion

We report that selatogrel inhibits fibrinogen binding to mouse platelets and blunts experimental thrombosis in guinea pigs only 10 min after the subcutaneous injection of selatogrel, confirming the rapid onset of anti-thrombotic activity. In addition, antithrombotic activity after the intravenous infusion or subcutaneous injection confirmed that the efficacy of selatogrel is independent of the route of administration. Furthermore, selatogrel induced the dissolution of the newly formed thrombus, but also consolidated platelet-rich thrombi in vivo, suggesting that P2Y12 receptor antagonism not only prevents thrombus growth, but also induces the dissolution of platelet-rich thrombi.

Classical light transmission aggregometry, following the method established by Born [24], requires blood samples of several millilitres which are then centrifuged to generate platelet-rich plasma. To obtain a sufficient volume of blood for aggregometry with rodent platelets, usually, one animal needs to be sacrificed for each time point. To avoid this process, we developed a novel platelet function test that enabled the monitoring of platelet function in a small volume of blood, and, thus, sampling of a series of timepoints. To avoid post-sampling anticoagulation in the test tube, heparin was infused into the animal prior to the start of the experiment. Furthermore, the binding of fluorescent-labelled fibrinogen to ADP-activated platelets was chosen as the functional readout to monitor the antagonism of the P2Y12 receptor. This experimental approach demonstrated that selatogrel has a rapid onset of action after subcutaneous injection into mice, with a ≥60% inhibition of fibrinogen binding after 10 min. To confirm that the inhibition of fibrinogen binding in anticoagulated blood in vitro translates into antithrombotic activity in vivo, selatogrel was characterised in a modified Folts thrombosis model in guinea pigs. Selatogrel blunted ongoing thrombosis approximately 10 min after subcutaneous injection. These data confirmed the antithrombotic activity of selatogrel in mice and guinea pigs and emphasised the rapid onset of action after subcutaneous applications.

In addition, using real-time intravital microscopy in mice, we demonstrated that selatogrel infused during ongoing platelet thrombus formation not only blunted ongoing thrombus growth, but also induced the dissolution of newly formed platelet-rich thrombi, suggesting that the beneficial effect of selatogrel is bipartite: the inhibition of thrombus growth and the induction of thrombus dissolution. Intravital microscopy in mice avoided anticoagulation and therefore confirmed that thrombus dissolution induced by a reversible P2Y12 receptor antagonist, previously reported in an ex vivo perfusion system [11], is not facilitated by anticoagulation.

Furthermore, our data demonstrated that P2Y12-receptor-mediated intraplatelet signalling is also important during the process of thrombus consolidation. In our system, fibrin generation continued during the process of thrombus consolidation in vivo. However, the presence of fibrin did not prevent the selatogrel-mediated dissolution of consolidated platelet-rich thrombi, suggesting that P2Y12-receptor-mediated intraplatelet signalling is a continuous process important for the long-term stabilisation of the interaction between GPIIb/IIIa and fibrin.

Our data disagree with previous reports where ticagrelor was not able to induce the disaggregation of 30-min-old platelet aggregates in vitro [29]. This difference suggests that the process of consolidation of platelet aggregates in anticoagulated blood in vitro involves different elements as compared to the process of consolidation in vivo.

To date, P2Y12-receptor-mediated stabilisation of the active conformation of GPIIb/IIIa has been monitored by a conformation-specific antibody (PAC-1) [30]. The data presented here confirm that the conformational changes detected by PAC-1 correlate with the in vivo binding of fibrin. Furthermore, the binding mode of fibrin binding to the GPIIb/IIIa receptor has been reported to differ from the binding mode of fibrinogen [31]. The observation that selatogrel dissolved platelet-rich thrombi in vitro and in vivo, involving fibrinogen-binding and fibrin-binding, respectively, suggested that preventing stabilisation of the active conformation of GPIIb/IIIa equally impacts both binding sites.

In our experimental system, the dissolution of platelet-rich thrombi was limited to disintegration of the occlusive part of the platelet thrombi, leaving small mural platelet aggregates on the surface of the blood vessel. Similar mural platelet aggregates were also described in P2Y12-receptor-deficient mice after laser injury, confirming that the formation of mural platelet aggregates is independent of P2Y12 receptor function [32]. In agreement with this, in previous studies, we have shown that the formation and the stability of mural platelet aggregates is independent of P2Y12 receptor functions and is not reduced by selatogrel, consequently covering the damaged surface of the vessel wall to function as haemostatic seals [21]. Specifically, we demonstrated that the observed off-target activities of other reversible P2Y12 receptor antagonists destabilises haemostatic seals and, thus, indirectly causes a complete dissolution of platelet thrombi [21]. In contrast, selatogrel did not destabilise haemostatic seals. Therefore, the identification of stable mural platelet aggregates, acting as haemostatic seals, was only possible due to the absence of off-target activities, i.e., due to the high selectivity of the P2Y12 receptor antagonist selatogrel [33].

Furthermore, our data emphasise the functional consequences of a previous description of the “core and shell” architecture of platelet thrombi [34,35]. The data presented here suggest that the structural description of an “inner core” corresponds to the functional description of haemostatic seals. We have previously shown that the stability of haemostatic seals is independent of P2Y12 receptor functions [21]. Furthermore, the structural description of an “outer shell” corresponds to the formation of an occlusive thrombus which depends on P2Y12 receptor function, which consequently can be dissolved by P2Y12 receptor antagonists. Therefore, the data presented here suggest that thrombus dissolution, without the disintegration of haemostatic seals, presents a unique advantage of the highly selective P2Y12 receptor antagonist selatogrel.

Our data, generated in experimental animal models, were translated towards human thrombosis using in vitro light transmission aggregation with human platelet-rich plasma. In agreement with previous reports describing light transmission aggregation or the ex vivo perfusion of human blood [11,12,13,29], selatogrel induced the disaggregation of ADP-induced platelet aggregates, confirming the rapid onset of anti-thrombotic and pro-dissolution effects of selatogrel with human platelets.

The preclinical data presented here were corroborated in a phase II trial, where a selatogrel regimen achieved more rapid and more complete P2Y12 inhibition in patients with AMI than what is achievable with the currently available oral drugs [36]. Therefore, selatogrel appears to incorporate the characteristics of a P2Y12 receptor antagonist, which seems optimal for self-applications in a pre-hospital setting in the case of acute myocardial infarction. The concept of the clinical use of selatogrel in patients presenting with AMI is currently being tested in a phase III study (ClinicalTrials.gov NCT03487445).

## 5. Conclusions

Our data suggest that the application of selatogrel in patients experiencing a thrombotic event will not only blunt ongoing thrombosis but might also contribute to the dissolution of pre-existing platelet-rich thrombi. In an acute thrombotic event, such effects might contribute to reducing ischemic occlusion, thus improving blood flow in the affected blood vessels. In combination with the rapid onset of platelet inhibition after subcutaneous application, selatogrel appears to possess an ideal profile for applications in patients presenting with acute thrombotic events.

## Figures and Tables

**Figure 1 jcm-10-05349-f001:**
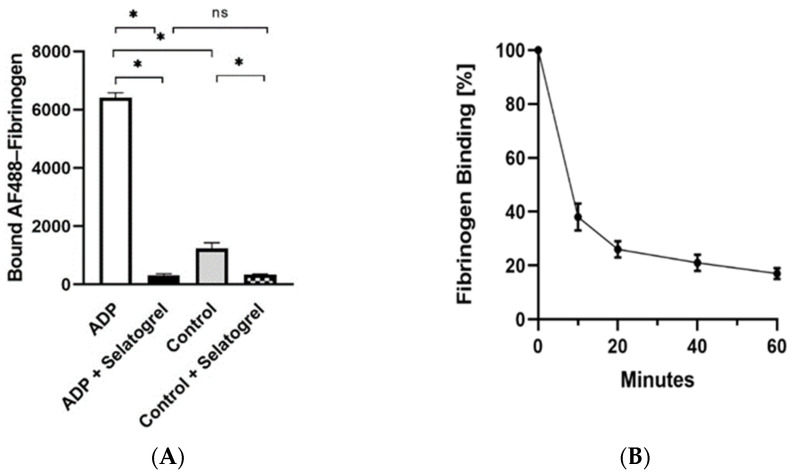
Selatogrel inhibits fibrinogen binding after subcutaneous applications in mice. Binding of FITC–fibrinogen to ADP-activated mouse platelets. Panel (**A**): Fibrinogen binding to platelets, expressed in arbitrary units as the mean fluorescence intensity (MFI) (average ± standard deviation). Asterisks indicate *p* < 0.001 in a Student’s t-test; ns, not significant. Panel (**B**): Time course of the inhibition of FITC–fibrinogen. Selatogrel (0.2 μg/kg) was injected subcutaneously in the lower abdomen of mice and blood samples were taken before injection (0) and 10, 20, 40, and 60 min after injection. Blood samples were processed as described in the Materials and Methods section. Maximal binding of FITC–fibrinogen in the absence of a P2Y12 receptor antagonist was defined as a 100% value. All other data are expressed relative to the 100% control. Fibrinogen binding (%) (y-axis) is presented relative to time (minutes) (x-axis). Error bars indicate the standard deviation (*n* = 3).

**Figure 2 jcm-10-05349-f002:**
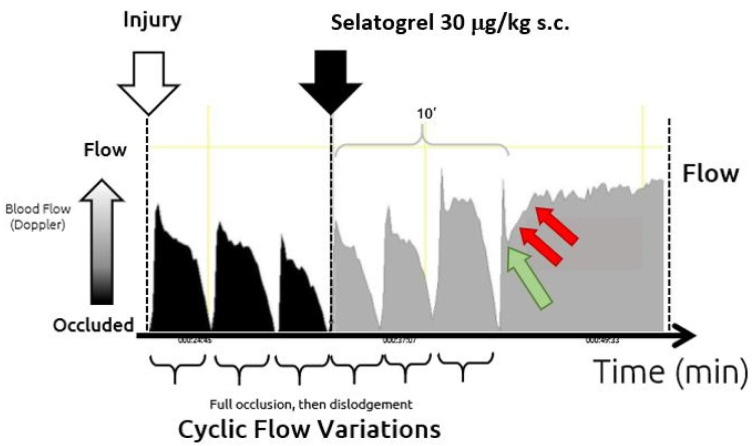
Modified Folts model in guinea pigs. Blood flow velocity (volts) (y-axis) is shown relative to time (minutes) (x-axis). Mechanical injury of the carotid artery is indicated by a white arrow. The black tracing indicates CFVs in the absence of antagonists. Selatogrel injection (30 μg/kg s.c.) is indicated by a black arrow. Subsequent CFVs are indicated by grey tracings. The green arrow indicates the initiation of antithrombotic activity, and the red arrows indicate the increase in blood flow velocity, suggestive of dissolution of the existing platelet thrombus after selatogrel application.

**Figure 3 jcm-10-05349-f003:**
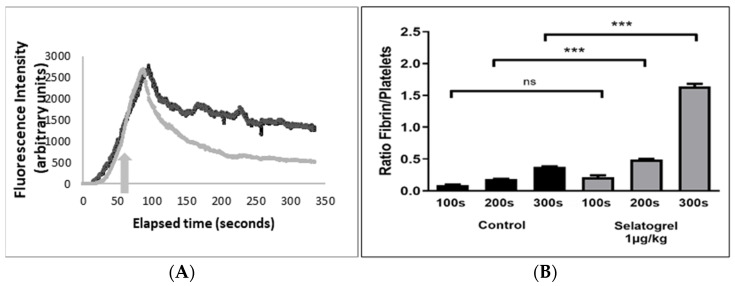
Intravital microscopy in mice. Laser-induced thrombus formation was monitored by real-time intravital microscopy. Panel (**A**): Increases in integrated fluorescence intensity (y-axis) are indicative of platelet incorporation into the growing thrombus and were monitored over time (x-axis). Black tracing represents the incorporation of platelets into a growing platelet-rich thrombus induced in the absence of an antagonist. The grey tracing represents the incorporation of platelets where selatogrel was injected 62 s after laser injury (1 μg/kg) (grey arrow). Panel (**B**): The incorporation of fibrin into growing platelet thrombi was quantified using a fluorescence-labelled anti-fibrin monoclonal antibody and expressed as the ratio of the fibrin signal relative to the observed platelet signal (y-axis). The fibrin/platelet signal ratio is presented at timepoints of 100, 200, and 300 s after laser injury. Black bars represent the absence of an antagonist, and grey bars represent the condition in the presence of selatogrel infusion (1 μg/kg). Brackets indicate a statistical comparison; ns, not significant, *** = *p* < 0.05.

**Figure 4 jcm-10-05349-f004:**
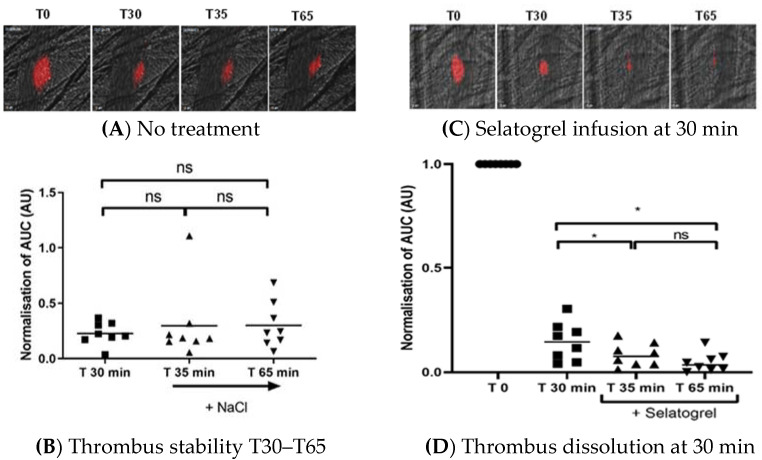
Dissolution of consolidated thrombi. Platelet thrombus formation was induced by laser injury. The stability of the platelet thrombi was monitored for up to 65 min. Panel (**A**): Platelet thrombus size in the absence of an antagonist. Representative images of platelet thrombi (red) at T0 (maximal size of the initial thrombus) and at 30 min (T30), 35 min (T35), and 65 min (T65). Panel (**B**): Area under the curve of the integrated fluorescence intensity normalised to the initial thrombus for T30, T35, and T65 (y-axis); ns, not significant. Panel (**C**): Representative images of platelet thrombi (red) at T0 (maximal size) and at 30 min (T30), 35 min (T35), and 65 min (T65). Selatogrel was infused at T30, and the thrombus sizes at T35 and T65 are shown. Panel (**D**): Area under the curve of the integrated fluorescence intensity normalised to the initial thrombus (T0) (y-axis). In the presence of selatogrel, the differences between integrated fluorescence intensity at T35 and T65 relative to T30 were statistically significant (* = *p* < 0.05) (ns, not significant).

**Figure 5 jcm-10-05349-f005:**
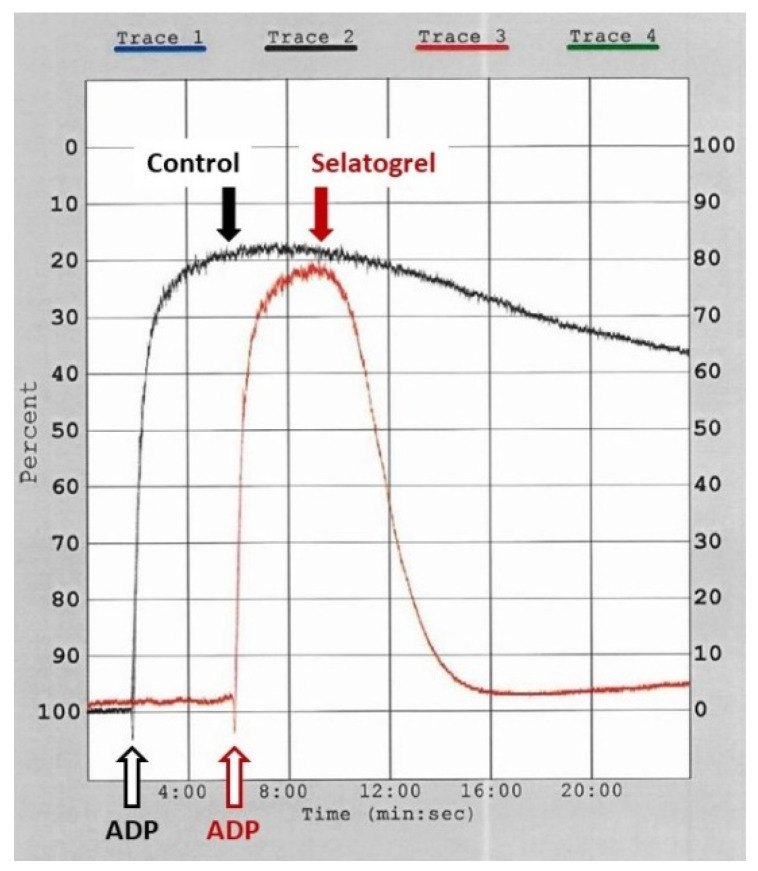
Dissolution of human platelet aggregates. Light transmission aggregation with human platelet-rich plasma containing physiological calcium concentrations. Light transmission is expressed as the percentage aggregation (y-axis, right hand side). Prior to the initiation of the experiment, the aggregometer was calibrated with platelet-poor plasma (100% aggregation) and platelet-rich plasma (0% aggregation). Aggregation was induced by the addition of ADP (10 μM), open arrows. Vehicle (Control) or selatogrel were injected at the peak aggregation (filled arrows, black = control, red = selatogrel 1 μM).

## Data Availability

The data presented in this study are available on request from the corresponding author. The data are not publicly available due to their storage in the proprietary electronic laboratory notebook of Idorsia Pharmaceuticals Ltd.

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
