# Peer review of "The P2Y12 Receptor Antagonist Selatogrel Dissolves Preformed Platelet Thrombi In Vivo"

_jcm, 2021, doi:10.3390/jcm10225349_

Round 1

Reviewer 1 Report

The manuscript by Crescence et al. demonstrate that ability of selatogrel to inhibit ongoing thrombosis along with its potential to dissolve pre-existing platelet-rich thrombi. The authors have used use multiple and innovative assays to conclusively investigate the anti-thrombotic effects of selatogrel. Overall, the study design is designed appropriately and provides important insight into the mode of action of selatogrel. Here are some comments.

  • Figure 1a: It is confusing what the open bar represents? Does it represent an unstimulated sample or ADP-stimulated control sample? It is mentioned in the text that “also in non-stimulated platelets, constitutive P2Y12 receptor signaling supported a baseline binding of AF488-fibrinogen of an average MFI of 1253”. If that is the case, where is the ADP stimulated control sample to compare it with Selatogrel-treated samples (3 uM) and stimulated with ADP?
  • Figure 1: Statistical analysis is not shown.
  • Statistical analysis should be done in figure 5 to make data meaningful.
  • While the authors have established that treatment with selatogrel inhibits the affinity of GPIIbIIIa towards fibrinogen binding, indicative of reduced inside-out signaling. They are trying to investigate the effects of seatogrel on consolidated thrombus, which is an important finding of this study. It is the outside-in signaling that plays a vital role in thrombus consolidation and stabilization. Therefore, it would be ideal to investigate the effects of selatogrel on outside-in signaling using a clot retraction assay and see if selatogrel can also prevent fibrin clot retraction.
  • Would it be possible to compare the effects of Selatogrel with other P2Y12 receptor antagonists such as Cangrelor or Ticagrelor? This can be performed in a manner similar to Figure 5.

Minor comments:

  • In the introduction, it is mentioned (Line no 33) that P2Y12 receptor is needed for the initiation of platelet aggregation. I believe that P2Y12 is required more for the amplification than initiation, which is mediated by GPVI receptor after stimulation with collagen.
  • It is unclear which experiments were performed on C57BL/6J and which ones on Balb/c mice. Was there a reason to specifically use female mice for this study?
  • Figure 5: The quality of the figure is extremely poor. Smaller arrow can be used to represent the addition of selatogrel. Color can be used to make a distinction between the two traces rather than mentioning/writing on the trace itself.

Author Response

Response to Reviewer 1

  • Figure 1a: It is confusing what the open bar represents? Does it represent an unstimulated sample or ADP-stimulated control sample? It is mentioned in the text that “also in non-stimulated platelets, constitutive P2Y12 receptor signaling supported a baseline binding of AF488-fibrinogen of an average MFI of 1253”. If that is the case, where is the ADP stimulated control sample to compare it with Selatogrel-treated samples (3 uM) and stimulated with ADP?
    Response:
    Figure 1A. has been revised as proposed and the text in the result section adapted..

  • Figure 1: Statistical analysis is not shown.
    Response:

Figure 1. A.  has been revised and statistical analysis was inserted.

  • Statistical analysis should be done in figure 5 to make data meaningful.
    Response:

We inserted a new Figure 5. with a clear color scheme.
Light transmission aggregation was performed with a Chrono-Log aggregometer.  While this aggregometer allows utmost experimental flexibility, it only delivers a graphical output in the form of aggregation tracings.  In contrast to other aggregometers, this data does not allow statistical analysis of aggregation slopes etc.  We inserted as much quantitative information as possible in the text of the result section.

  • While the authors have established that treatment with selatogrel inhibits the affinity of GPIIbIIIa towards fibrinogen binding, indicative of reduced inside-out signaling. They are trying to investigate the effects of selatogrel on consolidated thrombus, which is an important finding of this study. It is the outside-in signaling that plays a vital role in thrombus consolidation and stabilization. Therefore, it would be ideal to investigate the effects of selatogrel on outside-in signaling using a clot retraction assay and see if selatogrel can also prevent fibrin clot retraction.
    Response:
    We have done such experiments in which clot formation and subsequent retraction was stimulated by addition of 6 mM calcium into citrate-anticoagulated PRP. In comparison to vehicle, addition of selatogrel did not impair clot retraction. We propose that addition of calcium also enhances formation of thrombin which is then activating platelets via mechanisms which are independent of the P2Y12 receptor and thus, obscuring the potential influence of P2Y12 receptor signaling on outside-in signaling.

  • Would it be possible to compare the effects of Selatogrel with other P2Y12 receptor antagonists such as Cangrelor or Ticagrelor? This can be performed in a manner similar to Figure 5.
    Response:
    In similar experiments, ticagrelor also induced disaggregation of human platelet aggregates in vitro. However, with a slower effect (probably due to a slower on-rate).  However, since we did not profile ticagrelor in our experiments in vivo and not to deviate from the focus on selatogrel, we opted not to include this data in this manuscript.   

Minor comments:

  • In the introduction, it is mentioned (Line no 33) that P2Y12 receptor is needed for the initiation of platelet aggregation. I believe that P2Y12 is required more for the amplification than initiation, which is mediated by GPVI receptor after stimulation with collagen.
    Response:
    We agree with the reviewers comment and adapted the text to:
    " ADP-mediated activation of the P2Y12 receptor has been reported to be essential for the amplification of platelet aggregation and thus thrombus formation"

  • It is unclear which experiments were performed on C57BL/6J and which ones on Balb/c mice. Was there a reason to specifically use female mice for this study?
    Response:
    C57BL/6J were used for intravital microscopy experiments (Marseille, France) and Balb/C mice were used for the s.c. application of selaotogrel (Allschwil, Switzerland).
    This was now specified in the Materials and Methods section.
    To comply with the Swiss animal welfare regulations, the Balb/c mice were housed in groups. To avoid aggression between animals during group-housing, only female mice were used.

  • Figure 5: The quality of the figure is extremely poor. Smaller arrow can be used to represent the addition of selatogrel. Color can be used to make a distinction between the two traces rather than mentioning/writing on the trace itself.

Response:
A new revised Figure 5. with better quality and with a clear color pattern was inserted (see below).

Figure 5. Revised

Reviewer 2 Report

I was glad to review a preclinical study that demonstrates the use of selatogrel, a short acting intravenous p2y12 inhibitor on thrombus dissolution both in animal models and ex vivo. The authors are to congratulate on the extensive methodology description and results presentation. I have only minor comments to make:

  • Reduce the size of the Introduction section. You should keep only what’s important an not do an excessive review of the literature
  • In the last paragraph of the introduction provide concisely the aims of your study in the form of i), ii), iii).

Author Response

Response to Reviewer 2

I have only minor comments to make:

  • Reduce the size of the Introduction section. You should keep only what’s important an not do an excessive review of the literature
    Response:

The introduction section was shortened as recommended. Specifically, we removed the detailed description of the proteins involved in intra-platelet signaling between the P2Y12 receptor and GPIIb-IIIa.

  • In the last paragraph of the introduction provide concisely the aims of your study in the form of i), ii), iii).

Response:
We inserted the aims of our study in the last paragraph of the introduction:
Quote:
" The aims of our study was i) to characterize the rapid absorption of selatogrel after subcutaneous application to mice, ii) to characterize the antithrombotic effects of selatogrel, in the absence of anticoagulation, in experimental thrombosis models in vivo in mice and guinea pigs and iii) to investigate the potential effect of selatogrel on thrombus dissolution in vivo."